# Catch assemblages in the small-scale trap fishery with relation to hydrographic features of a tropical bay in the Gulf of Thailand

**Amonsak Sawusdee**[1☯], **Tanuspong Pokavanich**[2☯], **Sontaya Koolkalya**[3], **Jantira Rattanarat**[1], **Jenjira Kaewrat**[1], **Tuantong Jutagate**[4]*

1 School of Sciences, Walailak University, Nakhon Si Thammarat, Thailand, 2 Faculty of Fisheries, Kasetsart University, Bangkok, Thailand, 3 Faculty of Agricultural Technology, Rhambahibhannni Rajabhat University, Chantaburi, Thailand, 4 Faculty of Agriculture, Ubon Ratchathani University, Ubon Ratchathani, Thailand

☯ These authors contributed equally to this work.
* tuantong.j@ubu.ac.th

**Data Availability Statement:** All relevant data are within the paper and its Supporting Information files.

## Abstract

Catches from the small-scale trap fishery in Bandon Bay, Suratthani, Thailand, were monitored from 14 sites around the Bay, then disturbance to aquatic communities and catch assemblage were examined. At the same time, the hydrographical features of the bay were surveyed. The study was conducted throughout 2019 except in December, when a tropical cyclone made sampling impossible. In total, 17,373 animals from 118 species or species groups of aquatic animals were collected. The main target of the fishery, blue swimming crab *Portunus pelagicus*, contributed about 10% of the total catch in terms of number; meanwhile, another crab, *Charybdis affinis*, was the most dominant species (41% of total). W-statistics of Abundance-Biomass Comparison curves ranged between -0.025 and 0.031, indicating light disturbance to the communities in this fishing ground. The catch assemblage, based on number in catch composition, were divided into three main clusters and six subclusters by using the self-organizing map (SOM) technique. The SOM results showed that the catch assemblages differed based largely on temporal variation. The hydrographic features of Bandon Bay at times exhibited a layered structure and had strong spatial variation. The bay's current system was governed by motion of tidal currents; meanwhile, the circulation was governed by monsoonal wind and freshwater discharges. Tidal current was strong and ranged between approximately 0.6 m to 2.2 m. Water within the bay was always warmer than the outer sea. High water temperature was observed two times during the year: during monsoon transition 1 (April to May) and transition 2 (October). Salinity showed great spatial and temporal variation, differing by more than 5–10 ppt horizontally. It was possible to use these dynamic hydrological features of Bandon Bay to explain assemblage patterns of the trap-net catches.

**Funding:** The project received funding from the Agricultural Research Development Agency (www.arda.or.th): Project number: PRP6005010660 and PRP6105021960. The funders had no role in study design, data collection and analysis, decision to publish, or preparation of the manuscript.

**Competing interests:** The authors have declared that no competing interests exist

# Introduction

Production from marine capture fisheries in Thailand has ranged between 1.3 and 1.4 million tonnes since 2015 [1]. It is estimated that about 16% of this production was from artisanal small-scale fisheries (SSF) [2]. In 2020, there were about 57,000 registered fishing vessels in Thailand; among these, the ratio of small-scale fishing vessels (< 10 GT) to commercial ones (> 10 GT) was about 80:20, but the small-scale boats only accounted for about 15% of the total catch [3]. Although the fishing ground of SSF is limited to coastal areas, within 6 nautical miles from shore, Lymer et al. [4] reported that their catches of crabs, shrimps and shellfishes were higher by far than commercial fisheries, while the catches of cephalopods were roughly equal. Ferrer et al. [5] listed a number of existing challenges and vulnerabilities faced by the SSF sector in Southeast Asia, including poverty, market access, financial services and livelihoods. These problems, however, will not be exacerbated if the fishing ground for SSF (i.e., the coastal waters) remains intact and productive, and if aquatic animals are still diverse and abundant [5,6].

The coastal area is dynamic and highly productive, and large numbers of marine animal species coexist and thrive here, creating structured communities [7]. Communities of marine organisms in the coastal zone are also influenced by a range of abiotic factors, in particular salinity and temperature, and biotic processes, for example resource competition and life history traits, across multiple spatial and temporal scales, as well as by local circulation patterns [7–9]. In terms of spatial variation, differences can be observed along the longitudinal gradient from inshore to offshore [10,11]. Moreover, different habitat types also significantly influence the aquatic animals of the communities [9,12]. As for temporal variation, intra-annual variation is largely seasonal, while long-term variation depends on the level of stressors (e.g., over-fishing, pollution and climate change) from year to year [10,11]. These variations eventually affect the diversity, composition and assemblage patterns of the catch by fishing gears used in the area. Among the habitat types found in the coastal zone, bays, either semi-enclosed or open, act as the home of many fishes and other aquatic animals as a consequence of their mixture of features, which generally include mudflats connected to tidal and river channels, as well as a salinity gradient from freshwater to saline [7,10,13].

Bandon Bay (9° 20' 00" N, 99° 25' 00" E, Fig 1) is a shallow, open bay in the south of Thailand. Its area is 477 km$^2$, with 120 km of coastline and a mean depth of approximately 2 m. Offshore tides are moderate and are amplified toward the inner part of the bay. The tidal range at the middle of the bay is about 0.6 m during neap tide and 2.1 m during spring tide [14]. The bay is governed by Asian-Australian monsoonal climate, having the southwest monsoon (SWM) between July and September, and the northeast monsoon (NEM) between November and March. The period between April and May is the first monsoon transition (Transition 1), while the second monsoon transition (Transition 2) occurs in October. The average minimum air temperature is about 25°C between November and January, while the temperature can climb to highs of around 35°C between March and June. During the monsoon transitions, winds are weak (average speed <1 m/s), but substantially stronger during the monsoons (>3 m/s). Bandon Bay receives freshwater and nutrient inputs mainly from the Tapi River and also from 18 small river channels in the Tapi-Phumduang River system. High freshwater influxes to the bay via the rivers and by direct precipitation occur during the NEM [15]. The bay bottom has highly varied substrates of mud, clay and sand, and a large mudflat extends about 2 km offshore [16]. Due to the bay's characteristics, it is home of at least 340 fishes and other aquatic animals; mollusks and crustaceans are particularly abundant, and support many SSF activities [14,17]. Moreover, Bandon Bay also supports intensive sea farming, especially of bivalve species such as blood cockle, green mussel and oysters [18]. A recent report on SSF

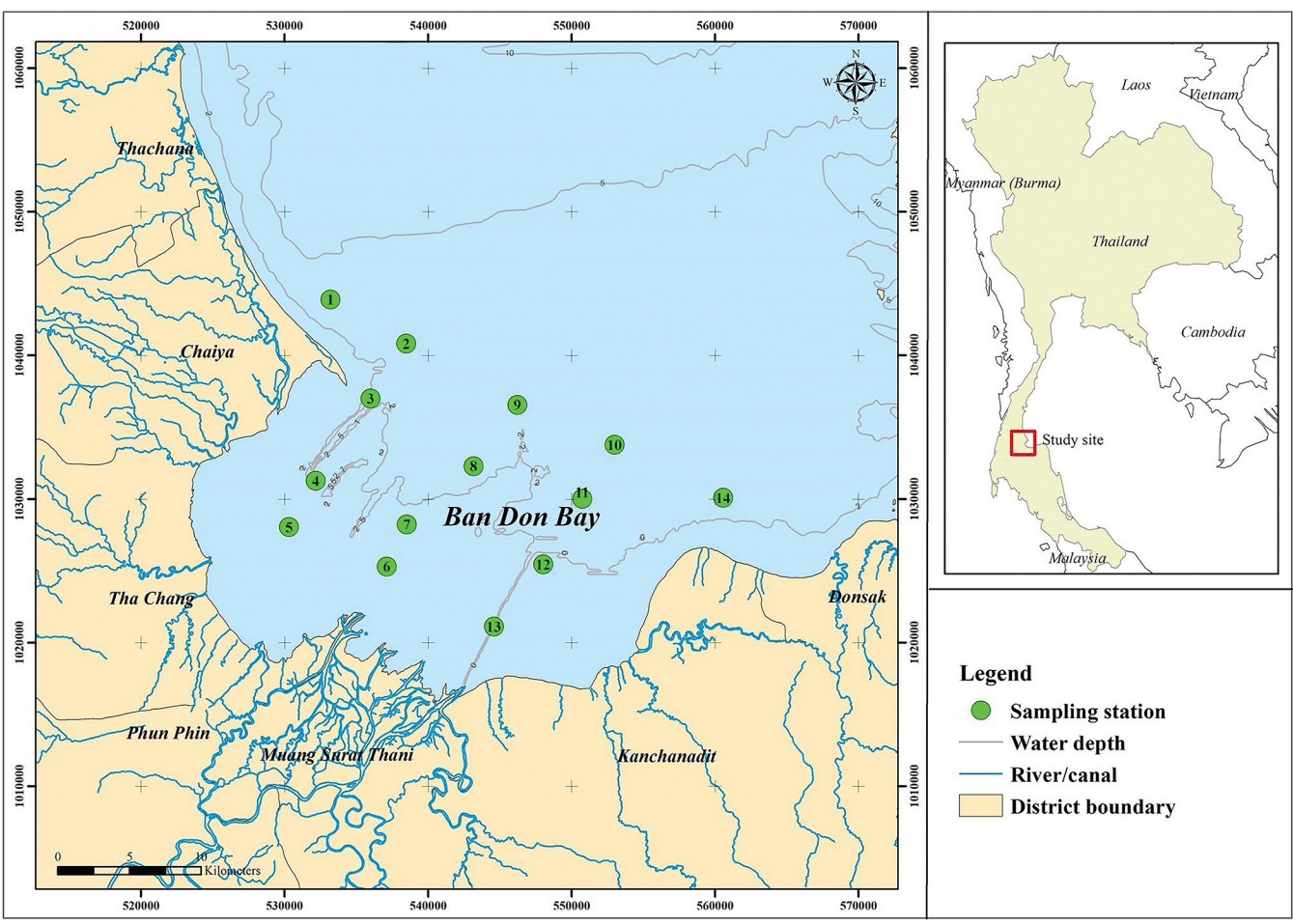

**Fig 1. Location of Bandon Bay and map of the designated sampling sites in this study.** Green dots indicate sampling sites, where fishers operated the trap fishery. Number within each dot is the site's identification.

revealed 2,890 registered fishers, and their estimated combined catch was about 450 tonnes in 2022 from almost 30 fishing gear types that target various types of aquatic animals [19].

Among the targeted species, the blue swimming crab (*Portunus pelagicus*) is considered as the most important species, and half of the country's production is from this Bay. The catch was 703 tonnes in 2021, which generated first-hand income for the fishers of over 130 million Thai Baht [1,14]. Operators using collapsible crab traps (hereafter "trap fishery") represent one of the major SSF in Bandon Bay, and are believed to employ the highest amount of fishing gear in this bay. The number of SSF trap fishers was estimated recently as roughly 10%, with the number of traps ranging between 500 and 1,000 traps per fisher. Meanwhile, the number of traps was as high 2,000–6,000 traps per boat for the commercial fisheries, which operated further offshore from the bay [20]. The traps used by SSF are made from 35×55×17 cm aluminum frames covered with 2.5-inch (6.4 cm) stretched-mesh netting; fish is used to bait the traps [14].

Despite the significance of the trap fisheries in Bandon Bay, little is known about its effects on the community of aquatic animals and patterns of catch assemblage. Previous studies have shown that the catch depends on where and when the traps are deployed. Moreover, the salient hydrographic features of the bay (i.e., water level, current, water temperature and salinity)

have not been sufficiently described within this bay, making their roles and effects on this fishery unclear, in particular at the scale of the SSF. The objectives of this study were, therefore, to (i) to assess the degree of disturbance by trap operation on aquatic animal communities in the fishing site, (ii) investigate the spatio-temporal variation of catch assemblage in the small-scale trap fishery, and (iii) understand the underlying water movements and changing physical properties of the water at different time-scales within Bandon Bay. The association of hydrographic features of the bay with catch assemblage and community structure in the trap fishing ground was also examined. These insights can be further applied to resource and fisheries management to sustain the small-scale trap fishery in Bandon Bay.

## Materials and methods

### Sampling sites and data collection

Fourteen (14) sites within the fishing ground of SSF in Bandon Bay were selected (Fig 1); once a month, 90 traps were taken from the fishers in each site. The traps were deployed for 12 h before retrieving. The sampling occurred from January to November 2018; in December, tropical cyclone Pabuk paused all SSF activity in Bandon Bay. The catches from each site were ice-packed and brought to Walailak University, 160 km from Bandon Bay. All animals were identified taxonomically, to species level if possible. Manuals used for identification were Nelson, Grande and Wilson [21] and FishBase (http://www.fishbase.org; [22]) for fishes, and Carpenter and Niem [23] and SeaLifeBase (http://www.sealifebase.org; [24]) for other aquatic animals. After identification and sorting, each species (or species group) was counted and weighed.

### Ethics statement

Ethical review and approval were not required for this study because data was based on a fishing survey of the trap fishers, and no live vertebrates or higher invertebrates were sacrificed for any experiments.

### Disturbance to community of aquatic animals

Degree of disturbance to the community of aquatic animals in the trap-fishing ground of Bandon Bay was assessed using the Abundance-Biomass Comparison (ABC) [25]. The ABC method compares the ranked distributions of abundance and biomass among species and presents the result as cumulative dominance (%) against species-rank. Abundance and weight of each species or group in each sampling event (i.e., month * site) were ranked and the cumulative percentage of both variables were plotted against its rank. The W-statistic, which is the numerical summarization of the ABC, measures differences in the overlap of the abundance and biomass curves, and is estimated as

$$W - statistic = \sum_{i=1}^{s} \frac{(B_i - A_i)}{[50(S-1)]} \tag{1}$$

where S is the number of species, $A_i$ is the abundance value of species with rank i, and $B_i$ is the biomass value of species with rank i [25]. The values of the W-statistic can range between -1 and +1; values near -1 indicate a highly stressed condition, values near zero show moderate stress, and values near 1 indicate a lack of stress [26]. Data analysis for ABC and the W-statistic was done by the software package "forams" [27]. The Kruskal–Wallis test was applied to test for statistical difference of the W-statistic among sampling sites and months, and Dunn's post-test was applied when the p-value at α = 0.05. Data analyses were done in R ((The R Foundation for Statistical Computing, Viennna, Austria, v.4.3.0).

## Assemblage patterns of the catches

Spatio-temporal patterns of catch assemblage were identified using an unsupervised artificial neural network "self-organizing map" (SOM, [28]). The SOM consists of a set of input units and output layers, formed by units arranged in a two-dimensional grid, connected with computational weights (i.e., weight vector). The SOM algorithm maps a set of input vectors (i.e., sampling event) onto a set of vectors of output units according to the characteristics of the input vector components (i.e., catch composition of each sampling event). Results of SOM are presented in the form of two-dimensional networks of neurons arranged on the map of a hexagonal lattice, in which sampling events with similar catch composition and weight of individual species are classified within the same or neighboring neurons [28,29]. During the training, the probability of occurrence (%O) of individual species in each cluster was also calculated, which was further used to describe the characteristics of each cluster. The number of output map units on the hexagonal lattice was estimated as $5\sqrt{n}$, where n = number of sampling events. The SOM toolbox was developed by the Laboratory of Computer and Information science, CIS, Helsinki University of Technology [30] and is available at https://github.com/ilariemimen/SOM-Toolbox. Hierarchical cluster analysis, by Ward's method, was used to examine the clusters on the SOM map by calculating the Euclidean distance between the weight vectors of each SOM unit [31]. Analysis of similarities (ANOSIM) was used to test statistical differences among clusters of catch assemblage pattern, by using %O, and was performed by R package "vegan" [32].

## Hydrodynamic modeling

A three-dimensional model of Bandon Bay was created using Delft3D-FLOW program [33]. The model realistically incorporated combined effects of wind, tides, Coriolis force, rivers, weather conditions and offshore salinity and water temperature on water movement within the bay. Wind and weather data are from *in-situ* measurements by a weather station with datalogger deployed on the roof of an aquaculture guard station situated in the middle of the bay. River discharge data are reported by the Royal Irrigation Department and were adjusted manually at some periods for the model to produce realistic salinity levels when compared with the measurements. The offshore tidal data were obtained from the OSU TPXO Tide model [34]. The estimates of offshore water temperature and salinity at various depths were derived from the in-house validated regional hydrodynamic model covering the whole Gulf of Thailand.

The model was calibrated and validated against spatial and temporal measured data from an intensive field observation program within the bay. Monitoring stations throughout the bay collected data at different monsoonal periods (i.e., NEM, Transition 1, SWM and Transition 2). Using a speed boat, multi-parameter sonde and current meter, 22 stations were measured within 6 hours during every monsoonal period. Validation of the model in terms of water level, water temperature and salinity was done using measured values from a station near Prab Island (9° 15.893'N, 99° 26.075'E) within the bay. The results showed good agreement between simulated and observed values, therefore the model was considered realistic enough to reproduce detailed hydrographic features of the bay (S1 Fig).

## Results

### Catches and disturbance to community

Over the entire survey period, 17,373 individuals with total weight of 461.3 kg from 118 species or species groups of aquatic animals were collected (Table 1). Fishes were the most diverse group with 48 species, followed by crabs (27), gastropods (11), echinoderms (11), mantis

**Table 1. Number (No), weight (W), and relative percentages (%No and %W) in the catch of taxa captured by the small-scale trap fishery in Bandon Bay between January and November 2018.**

| No | Name / Scientific name | Abbrev. | No | %No | W | %W |
|---|---|---|---|---|---|---|
| Anthrozoa | | | | | | |
| 1 | Sea anemone | sean | 23 | 0.13 | 67.9 | 0.01 |
| 2 | Sea pen | sepe | 26 | 0.15 | 395.5 | 0.09 |
| Gastropods | | | | | | |
| 3 | *Natica vitellus* | navi | 8 | 0.05 | 8.9 | 0.00 |
| 4 | *Pleuroploca* sp. | plsp | 7 | 0.04 | 77.3 | 0.02 |
| 5 | *Indothais* sp. | insp | 100 | 0.58 | 247.7 | 0.05 |
| 6 | *Lataxiena blosvillei* | labl | 2 | 0.01 | 14.7 | 0.00 |
| 7 | *Murex* sp.1 | musp1 | 17 | 0.10 | 114.0 | 0.02 |
| 8 | *Murex* sp.2 | musp2 | 15 | 0.09 | 62.2 | 0.01 |
| 9 | *Hemifusus* sp. | hesp | 6 | 0.03 | 233.1 | 0.05 |
| 10 | *Pugilina schumacher* | pusc | 3 | 0.02 | 99.3 | 0.02 |
| 11 | *Nassaria pusilla* | napu | 38 | 0.22 | 60.1 | 0.01 |
| 12 | *Nassarius siquijorensis* | nasi | 3 | 0.02 | 218.4 | 0.05 |
| 13 | *Cymbiola nobilis* | cyno | 3 | 0.02 | 379.1 | 0.08 |
| Bivalves | | | | | | |
| 14 | *Anadara inaequivalvis* | anin | 10 | 0.06 | 235.0 | 0.05 |
| 15 | *Tegillarca nodifera* | teno | 13 | 0.07 | 74.6 | 0.02 |
| 16 | *Mimachlamys sp.* | misp | 2 | 0.01 | 100.0 | 0.02 |
| 17 | *Paphia undulata* | paun | 3 | 0.02 | 11.1 | 0.00 |
| Cephalopods | | | | | | |
| 18 | *Sepia* sp.1 | sesp1 | 141 | 0.81 | 8,096.5 | 1.76 |
| 19 | *Sepia* sp.2 | sesp2 | 68 | 0.39 | 2,292.0 | 0.50 |
| 20 | *Sepiella inermis* | sein | 152 | 0.87 | 3,516.1 | 0.76 |
| 21 | *Octopus* sp. | ocsp | 6 | 0.03 | 639.2 | 0.14 |
| Horseshoe crabs | | | | | | |
| 22 | *Tachypleus gigas* | tagi | 14 | 0.08 | 2,150.7 | 0.47 |
| Mantis shrimps | | | | | | |
| 23 | *Harpiosquilla harpax* | haha | 3 | 0.02 | 6,013.3 | 1.30 |
| 24 | *Harpiosquilla raphidea* | hara | 3 | 0.02 | 2,754.8 | 0.60 |
| 25 | *Oratosquillina interrupta* | orin | 8 | 0.05 | 2,934.8 | 0.64 |
| 26 | *Oratosquilla woodmasoni* | orwo | 11 | 0.06 | 20.1 | 0.00 |
| 27 | *Oratosquilla nepa* | orne | 54 | 0.31 | 795.2 | 0.17 |
| Shrimps | | | | | | |
| 28 | *Macrobrachium rosenbergii* | maro | 31 | 0.18 | 116.0 | 0.03 |
| 29 | *Metapenaeus sp.* | mesp | 35 | 0.20 | 39.2 | 0.01 |
| 30 | *Penaeus semisulcatus* | pese | 2 | 0.01 | 98.9 | 0.02 |
| 31 | *Penaeus silasi* | pesi | 15 | 0.09 | 145.6 | 0.03 |
| Hermit crabs | | | | | | |
| 32 | *Diogenes* sp.1 | disp1 | 43 | 0.25 | 92.0 | 0.02 |
| 33 | *Diogenes* sp.2 | disp2 | 165 | 0.95 | 889.8 | 0.19 |
| 34 | *Clibanarius infraspinatus* | clin | 804 | 4.63 | 8,192.0 | 1.78 |
| 35 | *Dardanus lagopodes* | dala | 54 | 0.31 | 180.4 | 0.04 |
| Crabs | | | | | | |
| 36 | *Dorippe quadridens* | doqu | 1,919 | 11.05 | 19,128.1 | 4.15 |
| 37 | *Neodorippe callida* | neca | 4 | 0.02 | 16.9 | 0.00 |

*(Continued)*

**Table 1.** (Continued)

| No | Name / Scientific name | Abbrev. | No | %No | W | %W |
|---|---|---|---|---|---|---|
| 38 | *Myomenippe hardwickii* | myha | 86 | 0.50 | 8,456.5 | 1.83 |
| 39 | *Matuta planipes* | mapl | 40 | 0.23 | 1,245.6 | 0.27 |
| 40 | *Matuta victor* | mavi | 34 | 0.20 | 604.5 | 0.13 |
| 41 | *Doclea armata* | doar | 32 | 0.18 | 152.1 | 0.03 |
| 42 | *Doclea rissoni* | dori | 15 | 0.09 | 924.4 | 0.20 |
| 43 | *Doclea* sp. | dosp | 117 | 0.67 | 382.0 | 0.08 |
| 44 | *Doclea canalifera* | doca | 37 | 0.21 | 2,416.5 | 0.52 |
| 45 | *Charybdis affinis* | chaf | 7,195 | 41.41 | 131,080.1 | 28.41 |
| 46 | *Charybdis anisodon* | chan | 80 | 0.46 | 1,027.1 | 0.22 |
| 47 | *Charybdis feriata* | chfe | 108 | 0.62 | 13,501.2 | 2.93 |
| 48 | *Charybdis truncata* | chtr | 5 | 0.03 | 15.7 | 0.00 |
| 49 | *Lupocycloporus gracilimanus* | lugr | 5 | 0.03 | 34.9 | 0.01 |
| 50 | *Portunus haanii* | poha | 2 | 0.01 | 8.1 | 0.00 |
| 51 | *Portunus pelagicus* | pope | 1,673 | 9.63 | 118,744.2 | 25.74 |
| 52 | *Portunus sanguinolentus* | posa | 6 | 0.03 | 618.0 | 0.13 |
| 53 | *Scylla olivacea* | scol | 10 | 0.06 | 2,809.6 | 0.61 |
| 54 | *Thalamita crenata* | thcr | 2 | 0.01 | 639.1 | 0.14 |
| 55 | *Thalamita spinimana* | thsp | 131 | 0.75 | 4,074.0 | 0.88 |
| 56 | *Thalamita sima* | thsi | 17 | 0.10 | 698.4 | 0.15 |
| 57 | *Xiphonectes hastatoides* | xiha | 1 | 0.01 | 8.3 | 0.00 |
| 58 | *Galene bispinosa* | gabi | 6 | 0.03 | 591.8 | 0.13 |
| 59 | Unidentified crabs | crab | 1 | 0.01 | 873.8 | 0.19 |
| 60 | *Halimede ochtodes* | haoc | 22 | 0.13 | 127.4 | 0.03 |
| 61 | *Seulocia vittata* | sevi | 27 | 0.16 | 12.0 | 0.00 |
| 62 | *Varuna yui* | vayu | 1 | 0.01 | 286.0 | 0.06 |
| Starfishes | | | | | | |
| 63 | *Ophiocnemis* sp. | opsp | 2 | 0.01 | 0.4 | 0.00 |
| 64 | *Luidia* sp. | lusp | 73 | 0.42 | 1,079.0 | 0.23 |
| 65 | Sea Star 1 | sest1 | 11 | 0.06 | 28.1 | 0.01 |
| 66 | Sea star 2 | sest2 | 567 | 3.26 | 2,061.6 | 0.45 |
| Sea urchins | | | | | | |
| 67 | *Temnopleurus toreumaticus* | teto | 1,460 | 8.40 | 7,509.3 | 1.63 |
| 68 | *Arachnoides placenta* | arpl | 5 | 0.03 | 47.9 | 0.01 |
| Sea cucumbers | | | | | | |
| 69 | *Acaudina* sp.1 | acsp1 | 155 | 0.89 | 2,459.6 | 0.53 |
| 70 | *Acaudina* sp.2 | acsp2 | 41 | 0.24 | 389.1 | 0.08 |
| 71 | *Phyllophorella kohkutiensis* | phko | 63 | 0.36 | 853.8 | 0.19 |
| 72 | Sea cucumber | secu | 2 | 0.01 | 1,620.2 | 0.35 |
| Fishes | | | | | | |
| 73 | *Muraenesox cinereus* | muci | 1 | 0.01 | 122.9 | 0.03 |
| 74 | *Sardinella gibbosa* | sagi | 5 | 0.03 | 43.6 | 0.01 |
| 75 | *Thryssa kammalensis* | thka | 3 | 0.02 | 14.4 | 0.00 |
| 76 | *Hexanematichthys sagor* | hesa | 1 | 0.01 | 22.6 | 0.00 |
| 77 | *Batrachomoeus trispinosus* | batr | 25 | 0.14 | 2,302.5 | 0.50 |
| 78 | *Platycephalus indicus* | plin | 5 | 0.03 | 291.3 | 0.06 |
| 79 | *Vespicula trachinoides* | vetr | 29 | 0.17 | 138.2 | 0.03 |
| 80 | *Ambassis vachellii* | amva | 4 | 0.02 | 33.9 | 0.01 |

(*Continued*)

**Table 1.** (Continued)

| No | Name / Scientific name | Abbrev. | No | %No | W | %W |
|---|---|---|---|---|---|---|
| 81 | *Ostorhinchus fasciatus* | osfa | 4 | 0.02 | 60.1 | 0.01 |
| 82 | *Petroscirtes* sp. | pesp | 1 | 0.01 | 9.3 | 0.00 |
| 83 | *Acentrogobius caninus* | acca | 1 | 0.01 | 31.9 | 0.01 |
| 84 | *Alepes djedaba* | aldj | 30 | 0.17 | 189.4 | 0.04 |
| 85 | *Carangoides praeustus* | capr | 2 | 0.01 | 44.0 | 0.01 |
| 86 | *Carangoides* sp. | casp | 1 | 0.01 | 2.5 | 0.00 |
| 87 | *Gazza minuta* | gami | 1 | 0.01 | 15.3 | 0.00 |
| 88 | *Nuchequula gerreoides* | nuge | 28 | 0.16 | 135.7 | 0.03 |
| 89 | *Secutor hanedai* | seha | 2 | 0.01 | 4.0 | 0.00 |
| 90 | *Lutjanus russelli* | luru | 1 | 0.01 | 45.0 | 0.01 |
| 91 | *Pomadasys maculatus* | poma | 1 | 0.01 | 12.4 | 0.00 |
| 92 | *Pomadasys kaakan* | poka | 5 | 0.03 | 57.8 | 0.01 |
| 93 | *Johnius amblycephalus* | joam | 7 | 0.04 | 126.0 | 0.03 |
| 94 | *Pseudosciaena soldado* | psso | 15 | 0.09 | 359.4 | 0.08 |
| 95 | *Otolithes ruber* | otru | 2 | 0.01 | 12.4 | 0.00 |
| 96 | *Pennahia anea* | pean | 4 | 0.02 | 23.3 | 0.01 |
| 97 | *Upeneus sulphureus* | upsu1 | 13 | 0.07 | 384.6 | 0.08 |
| 98 | *Upeneus sundaicus* | upsu2 | 49 | 0.28 | 1,485.2 | 0.32 |
| 99 | *Terapon jarbua* | teja | 152 | 0.87 | 2,154.1 | 0.47 |
| 100 | *Terapon puta* | tepu | 7 | 0.04 | 121.9 | 0.03 |
| 101 | *Terapon theraps* | teth | 53 | 0.31 | 386.6 | 0.08 |
| 102 | *Sphyraena jello* | spje | 1 | 0.01 | 6.7 | 0.00 |
| 103 | *Scatophagus argus* | scar | 1 | 0.01 | 29.3 | 0.01 |
| 104 | *Epinephelus coioides* | epco | 1 | 0.01 | 105.3 | 0.02 |
| 105 | *Epinephelus sexfasciatus* | epse | 22 | 0.13 | 652.6 | 0.14 |
| 106 | *Siganus canaliculatus* | sica | 58 | 0.33 | 1,298.4 | 0.28 |
| 107 | *Siganus javus* | sija | 37 | 0.21 | 1,443.7 | 0.31 |
| 108 | *Brachirus orientalis* | bror | 50 | 0.29 | 1,879.5 | 0.41 |
| 109 | *Brachirus harmandi* | brha | 14 | 0.08 | 335.7 | 0.07 |
| 110 | *Cynoglossus arel* | cyar | 3 | 0.02 | 34.5 | 0.01 |
| 111 | *Cynoglossus trulla* | cytr | 3 | 0.02 | 47.1 | 0.01 |
| 112 | *Cynoglossus* sp. 1 | cysp1 | 8 | 0.05 | 190.6 | 0.04 |
| 113 | *Cynoglossus* sp.2 | cysp2 | 14 | 0.08 | 192.8 | 0.04 |
| 114 | *Triacanthus nieuhofii* | trni | 4 | 0.02 | 139.2 | 0.03 |
| 115 | *Paramonacanthus choirocephalus* | pach | 65 | 0.37 | 1,005.7 | 0.22 |
| 116 | *Chelonodon* sp. | chsp | 22 | 0.13 | 2,703.7 | 0.59 |
| 117 | *Lagocephalus lunaris* | lalu | 58 | 0.33 | 1,690.2 | 0.37 |
| 118 | *Takifugu oblongus* | taob | 717 | 4.13 | 75,054.8 | 16.27 |
| | **Total** | | **17,373** | **100.00** | **461,330.9** | **100.00** |

shrimps (5), cephalopods (4), shrimps (4), hermit crabs (4), bivalves (4), horseshoe crabs (2) and coelenterates (1). Crabs were dominant in terms of number of individuals captured, and led by *Charybdis affinis* (7,195), *Dorippe quadridens* (1,919) and *P. pelagicus* (1,673). Sea urchin *Temnopleurus toreumaticus* (1,460) was also represented by more than 1,000 individuals overall during the survey. Several fishes, mostly demersal, were retained in the traps such as *Takifugu oblongus* (717), *Terapon jarbua* (151) and *Paramonacanthus choirocephalus* (65). Shrimps and hermit crabs also contributed in substantial number to the catches. Weight of individual

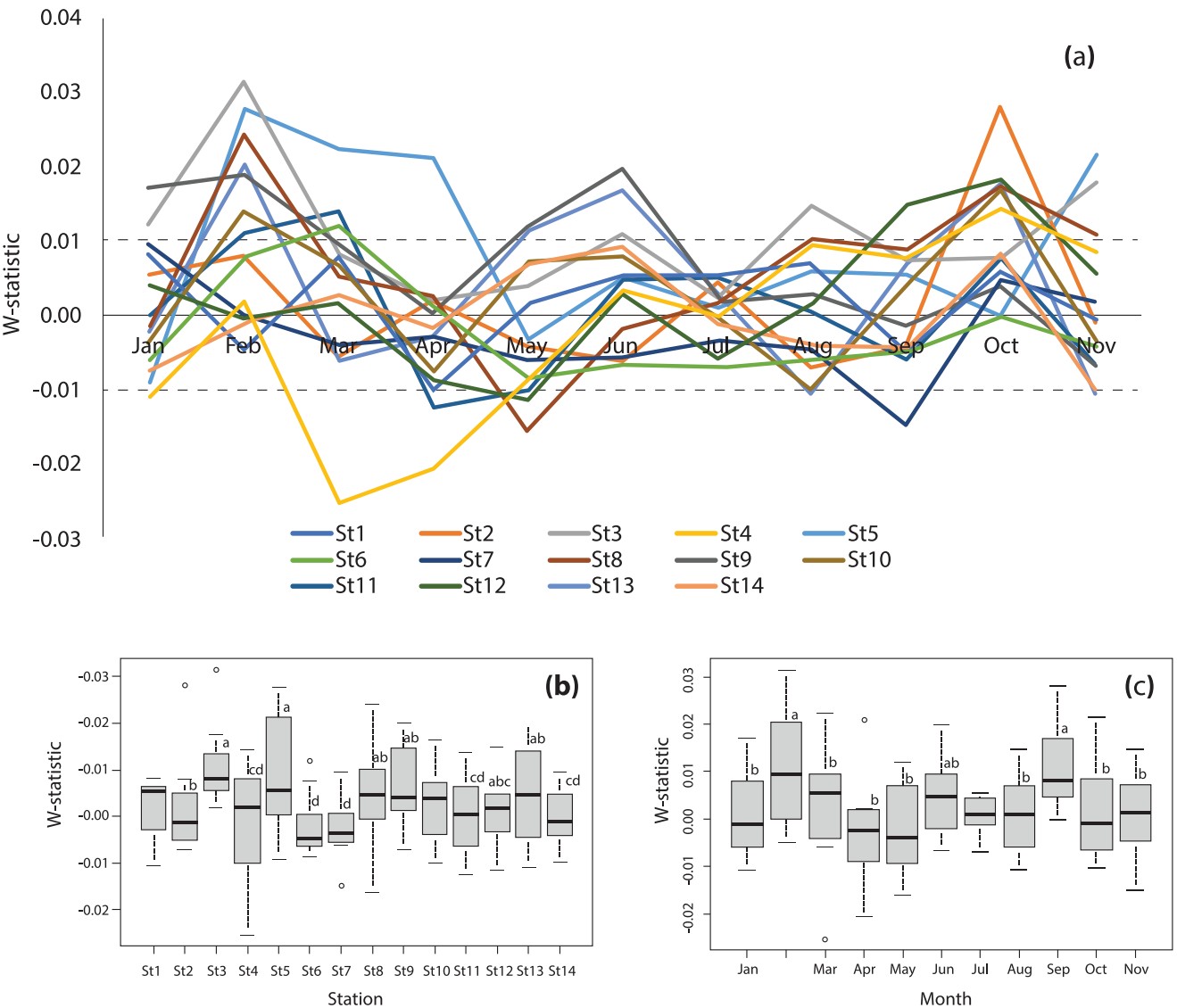

**Fig 2.** Estimated W-statistic values for aquatic communities in Bandon Bay (a) Temporal fluctuation of W-statistic values in each sampling site during study period, and boxplots of the W-statistic values among sites (b) and months (c). Different letters in (b) and (c) indicate statistically significant differences (P-value < 0.05) of the mean W-statistic values.

taxa was generally aligned with their quantity in the catch i.e., the higher the number of individuals, the higher their proportion by weight in the catch. Exceptions to this pattern include cephalopods, horseshoe crabs and mantis shrimps, for which individuals captured were relatively large

Temporal fluctuation at each sampling site produced W-statistic values that were between -0.025 and 0.031 (Fig 2A); examples of the ABC graph are shown in S2 Fig. This suggests that the communities of aquatic animals in the trap-fishing grounds of Bandon Bay were moderately disturbed. Although there was no clear spatial or temporal pattern in the W-statistic values, a significant difference was found both among sites (P = 0.042) and among months (P = 0.002) of sampling (Fig 2B and 2C). Sites 6 and 7 were highly stressed, and experienced

significantly higher interference than the other sites. Meanwhile, the community structure in the remaining sites was more stable, with only moderate disturbance. Temporal change of the W-statistic clearly revealed that community structure became more stable in February and October. In contrast, communities were most disturbed between April and May.

## Assemblage patterns of the catch

The SOM result was portrayed as 64 (i.e., 8 x 8) map units containing 154 sampling events, in which the sampling events with high similarity of species occurrence were grouped into the same or nearby map units. The final quantization and topographical errors of the SOM were 1.048 and 0.006, respectively, which were low enough to make the map reliable. The resulting hierarchical cluster analysis suggested three main clusters, each of which was further split into two sub-clusters (Fig 3A) that can be seen as map units in Fig 3B. Sampling events included in each map unit are presented in Fig 3C. Cluster A of the SOM can be described as a typical assemblage of the catch, since it contained 97 out of 154 sampling events, including all sampling events in July and almost all sampling events in January, May, June, August, September and November, as well as five in March and four in April. Cluster A was further split into A1 and A2, of which cluster A2 comprised most of the sampling events on the east side of the bay in July, September and December. Cluster B represents the assemblage pattern of catches from all sampling events in October and a few sampling events in November (3), April (2) and one each in January, February and June; sampling events in October and January were further split to Cluster B2. Most of the members in Cluster C were sampling events between February and April as well as two (2) sampling events in June and one each from May, August and

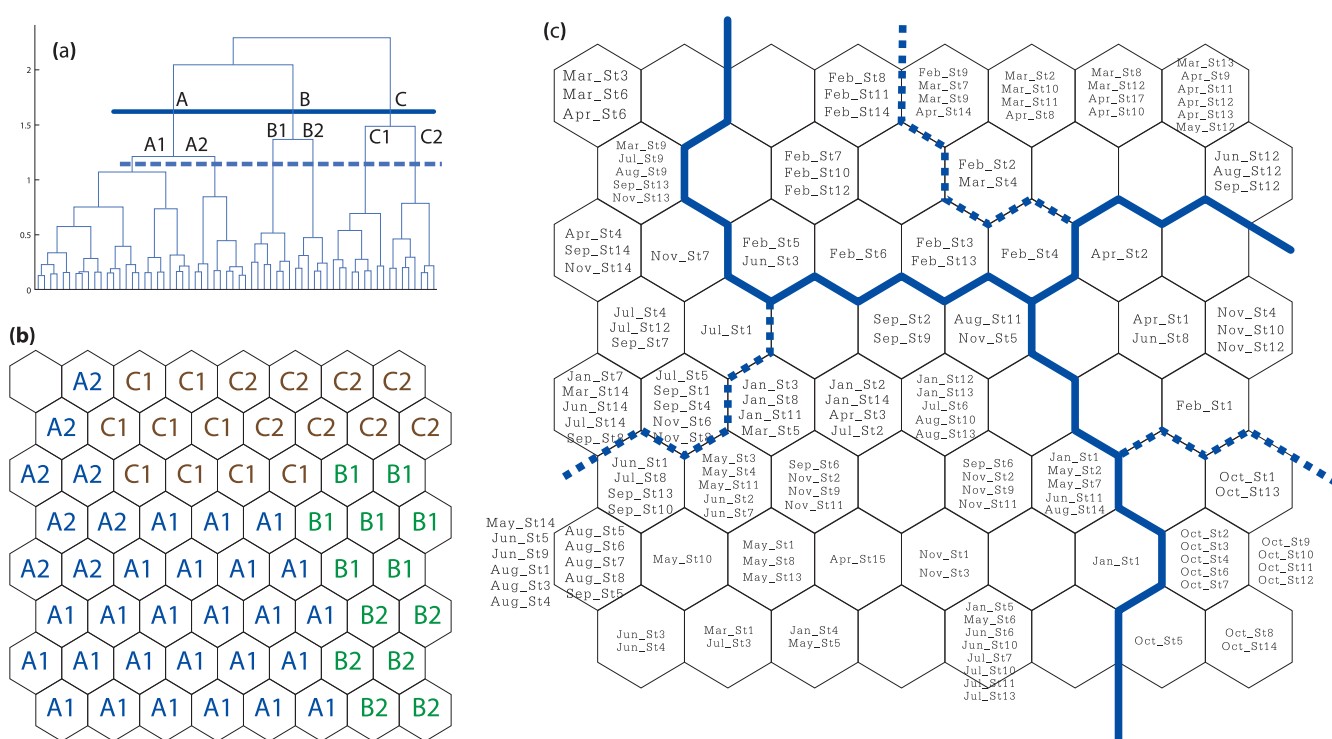

**Fig 3.** Distribution pattern of survey samples in the self-organizing map (SOM) cells: (a) dendrogram of SOM cells using Ward linkage method from Euclidian distance Matrix, (b) similarity of neighboring cells were used to group samples in clusters and sub-clusters, and (c) SOM showing clusters (bold line) and sub-clusters (dashed line).

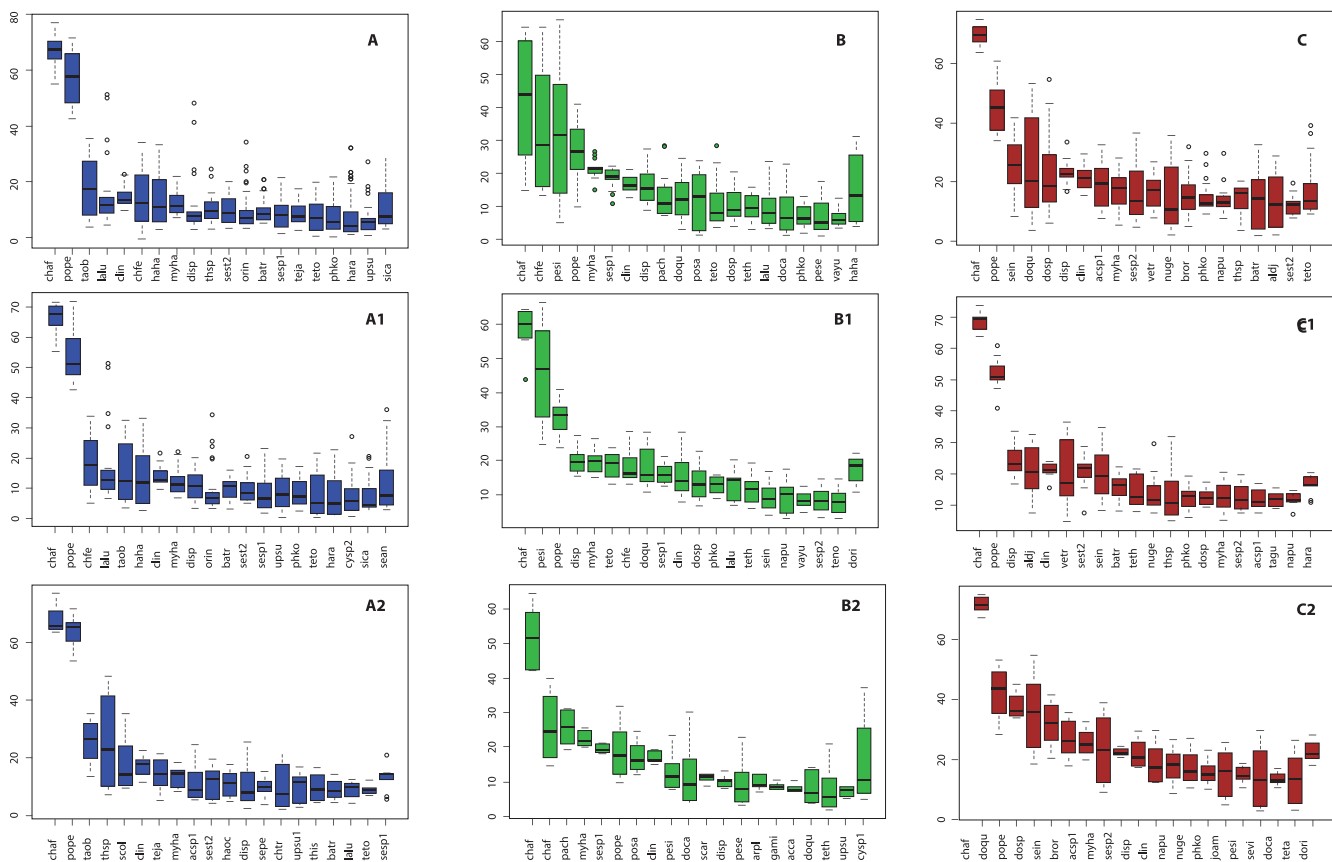

**Fig 4. Box plots showing occurrence probability (%) of each species (see full names in Table 1) in each cluster.** Values were obtained from the weight of virtual vectors of the trained SOM. The blue, orange and green columns indicate cluster A, B and C, respectively.

September. The catch assemblages of almost all sampling events in February were further split into Cluster C1, meanwhile Cluster C2 was dominated by the sampling events in March and April. The ANOSIM results confirmed that the similarity of assemblages of the catch within individual clusters, both among the three main clusters (P < 0.01) and among the six sub-clusters (P < 0.01).

Each main cluster and sub-cluster were characterized by the probability that a given species would occur within a catch assemblage. The assemblage pattern of cluster A (Fig 4A) was characterized by highly probability of occurrence (%O) (i.e., over 50%) of two crab species, namely *C. affinis* and *P. pelagicus*. The %O of the other species in Cluster A were all less than 20%, and only nine species represented over 10%; these included fishes (*Takifugu oblongus* and *Lagocephalus lunaris*), mantis shrimps (*Harpiosquilla harpax*) and crabs (*Charybdis feriata* and *Thalamita spinimana*). There were 30 species with %O more than 5% in cluster A. Species composition and %O values in sub-cluster A1 were quite similar to cluster A, meanwhile the %O of *T. oblongus* and *T. spinimana* were increased to over 20% in cluster A2 (Fig 4B and 4C). The %O of *P. pelagicus* was substantially lower in cluster B than cluster A, while crab *C. affinis* remained the species with highest %O in this cluster, though the value was lower than in clusters A and C (Fig 4D). The %O of species such as shrimp (*Penaeus silasi* and *Penaeus semisulcatus*), cephalopods (*Sepia* spp.) and other crabs were higher in cluster B than cluster A, and 33 taxa had %O above 5%. The obvious difference between sub-clusters B1 and B2 was the substantially higher %O of *P. silasi* in cluster B1 (Fig 4E and 4F). Moreover, there was substantially

lower %O of *P. pelagicus* and greater %O of *Sepia* spp. in cluster B2. Cluster C contained 58 species that had %O higher than 5%. As with the other two main clusters, the dominant species in cluster C were *C. affinis* and *P. pelagicus* (Fig 4G). Other high-%O species in this cluster included cephalopods (*Sepiella inermis* and *Sepia* spp.), crabs (*Dorippe quadridens* and *Doclea* spp.), hermit crabs (*Diogenes* spp. and *Clibanarius infraspinatus*) and fishes (*Vespicula trachinoides*, *Nuchequula gerreoides* and *Brachirus orientalis*). One difference between the two sub-clusters was that while species with high %O were included from various groups in cluster C1, crabs dominated cluster C2 (Fig 4H and 4I).

## Hydrographic features of Bandon Bay

Results reported here show that despite the bay's moderate size and lack of deep water in relation to its large tidal range, the hydrographic features at times exhibit a layered structure and have strong spatial variation. The shallow nature and large tidal range of Bandon Bay cause its circulation to be governed by the motion of tidal currents. The rising and falling tide in the Gulf of Thailand controls offshore water levels. Tides in the bay is mixed tide prevailing semi-diurnal and the Formzahl number, i.e., the division of the amplitude of the main single tidal constant from the main double tidal constant amplitude, is 1.37. Tidal range varies between approximately 0.6 m and 2.2 m. Tidal currents are strong (>60cm/s) during flood and ebb tides for the incoming and outgoing flow (Fig 5). Water movement is sluggish during the high and low tide. These strong tidal currents can stir fine bottom sediment, making the bay's water more turbid than the outer sea. The lowest sea level is observed in the middle of the year. The highest sea level occurs at the end of the year, from the combined effects of the seasonal sea level of the South China Sea system [35] and higher freshwater fluxes from the Tapi River system.

While the tidal current is strong, mean current (circulation) of Bandon Bay is governed by monsoonal wind and freshwater discharges. Fig 6 shows monthly mean water temperature, salinity and current pattern data from a numerical model (see the results of all months in S3–S5 Figs). The data show significant monsoonal variation, in which the degree of change depends greatly on location within the bay. In general, mean water temperature varies between 28 and 32°C, and And is generally homogeneous throughout the water column (Fig 7A and 7B, and S6–S8 Figs). Water within the bay is always warmer than the outer sea. Relatively higher water temperatures were observed two times during the year: at monsoon Transition 1 and Transition 2. Salinity has great spatial and temporal variation. During the NEM, salinity in the bay can differ by more than 5–10 ppt horizontally. Salinity is lowest near river month, especially during the wet season (NEM). During Transition 2 and the NEM, vertical structuring of salinity occurs in the eastern half of the bay, especially near the river mouth. This means that the tide cannot mix the water column thoroughly at all times, particularly during the neap tide. Compared to the eastern half of the bay (i.e., Prab Island), salinity levels in the western half (i.e., Sed Island) are more stable and have less vertical structure. Considering the mean flow velocity both along and across the bay's mouth (Fig 7C and 7D), results revealed that although the water column is well mixed by the tide, hydrographic conditions in Bandon Bay at times of the year exhibit strong three-dimensional features. During Transition 2 and the NEM, buoyancy forces of the near-surface fresher waters provided by the freshwater influxes are larger than the tidal mixing forces. This creates a two-layer current system, in which near-surface currents flow out of the bay to the northwest, while sub-surface currents flow in the opposite direction, to the east and toward the inner part of the bay. This flow pattern allows import of the cooler and more saline water and other water-borne material from the Gulf of Thailand to flow into the inner bay.

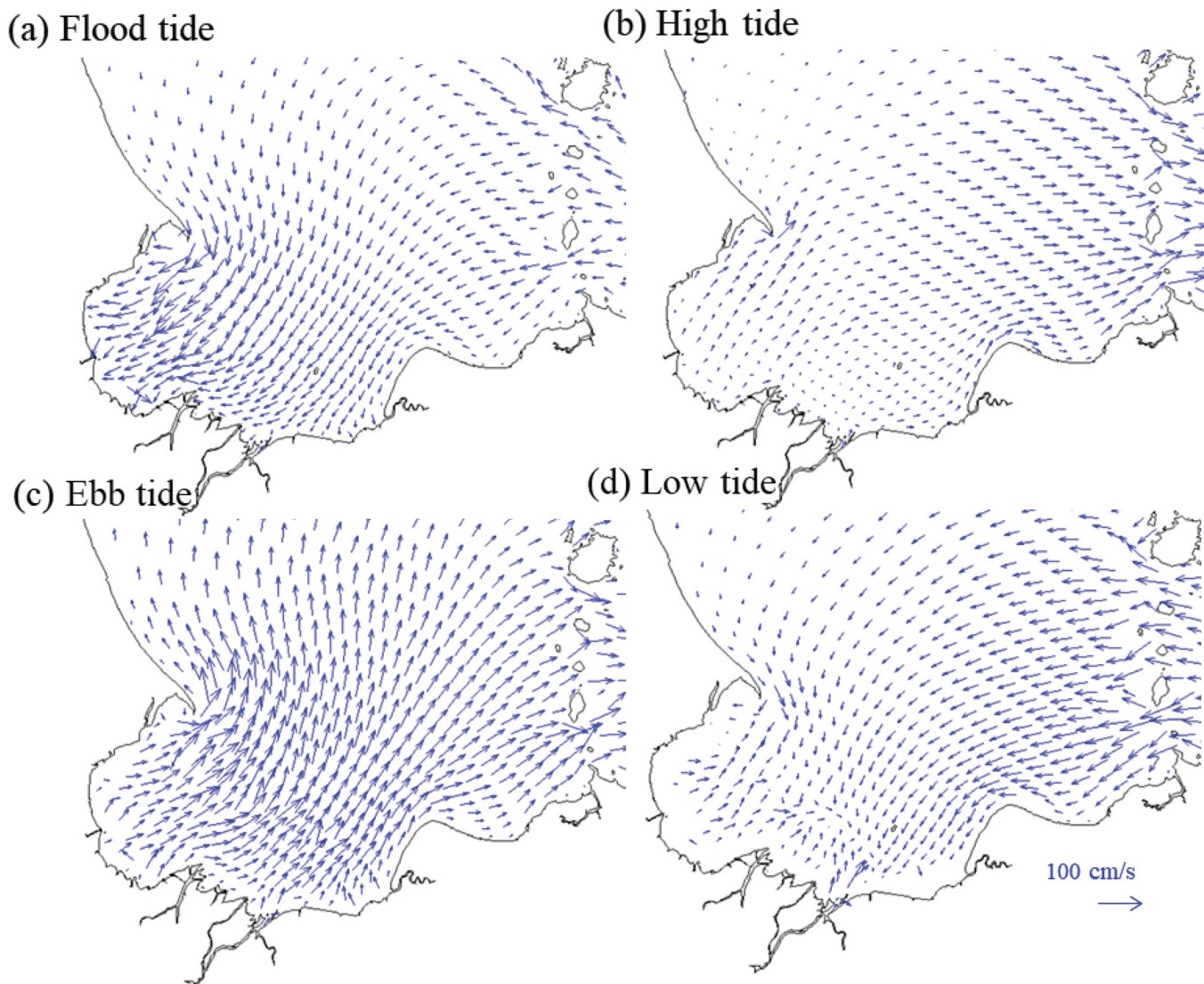

**Fig 5. Model of strong tidal current pattern at different tidal phases with the maximum current speed >60 cm/s that governs diurnal water movement and water column mixing within Bandon Bay.**

## Discussion

Although the small-scale trap fishery in Bandon Bay targets the blue swimming crab, this fishery employs indiscriminate fishing gear, similar studies of the bay have reported that these trap nets catch roughly 100 species of aquatic animals [14,17]. However, the catch composition from bottom gill nets, another common fishing gear for blue swimming crab, usually includes more species than traps—as many as 150 taxa [36]. Comparing the two gears, the traps tend to capture a greater number of crab species, whereas the number of fish species is higher in gill-nets [14]. Among the species caught, only one taxon exhibits true diadromy, the giant prawn *Macrobrachium soenbergii*. The remaining taxa are amphidromous species or marine visitors, entering the bay mostly for feeding [12,37,38]. Soe et al. [39] reported that the fishes of the inner bay community tend to be plankton feeders, meanwhile the community in the outer bay is dominated by piscivores. The two most dominant fish species caught by the trap fishery in

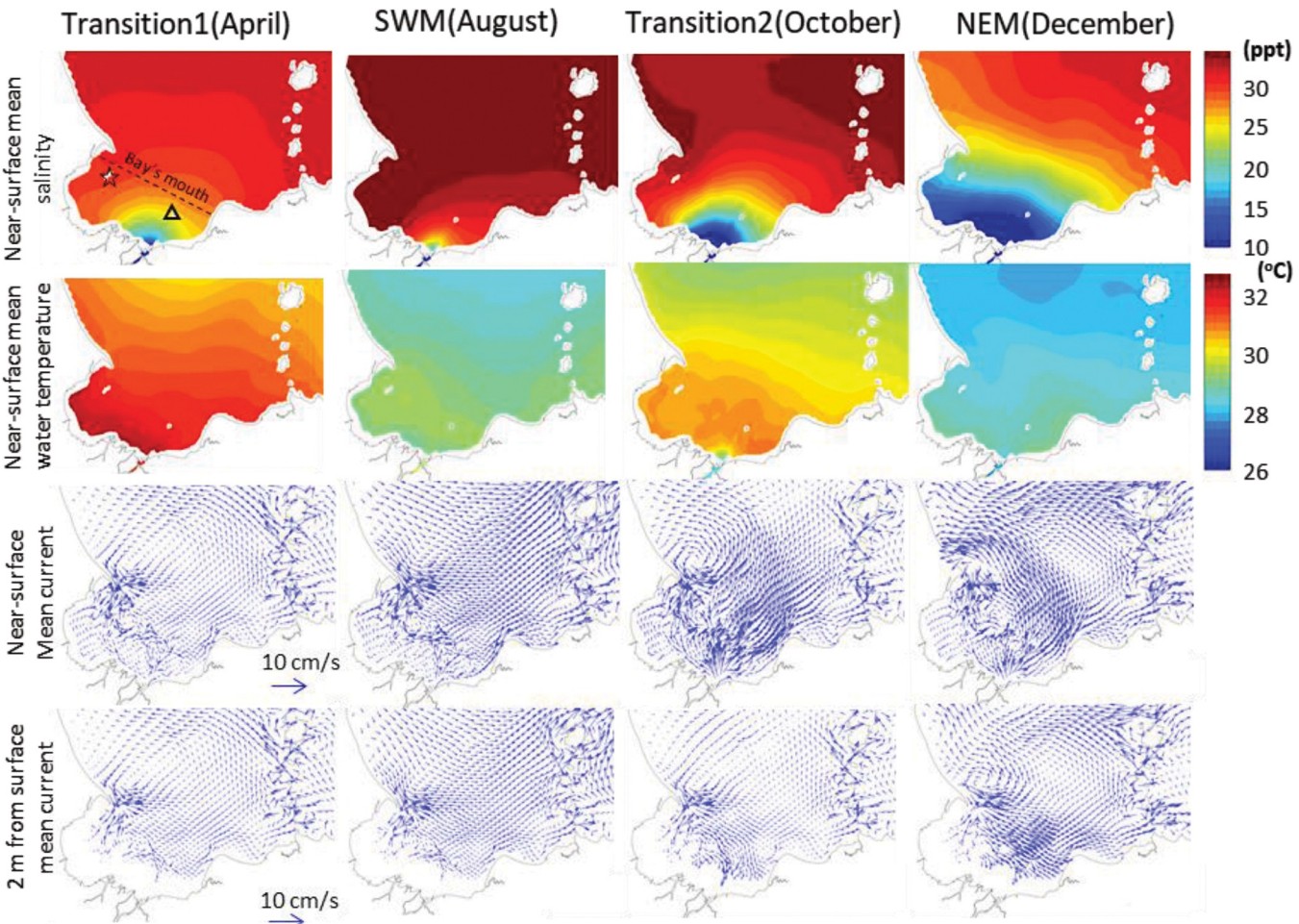

**Fig 6. Monthly mean current and physical properties of water of Bandon Bay from model at different monsoonal periods.** Dashed line, star and triangle indicate location of the bay's mouth, Sed Island and Prab Island, respectively.

Bandon Bay, namely *T. oblongus* and *T. jarbua*, are brackish to marine carnivores [22] that may enter traps to predate on other species already captured [40]. Dominance of *Charybdis* spp. in the catches, in particular *C. affinis*, reflected their abundance in the bay as well as their ability to adjust to salinity fluctuation, which could be between 10 and >30 ppt. Occhi et al. [41] revealed that *Charybdis* spp. are euryhaline, and can survive at salinity from 10 to 40 ppt. The amount of unwanted invertebrate bycatch observed in this fishery was similar to that of other fishing gears operated in Bandon Bay [17]. The fishers using traps in Bandon Bay commonly deal with the bycatch species by releasing them back to the sea, in which most of them are alive [17,20].

## Hydrographic features in relation to condition of the catch and assemblage patterns

The hydrological environment of the bay is very dynamic, particularly when contrasted with freshwater and coastal ecosystems, and such conditions may affect the condition and composition of the livings in different area within the bay [42,43]. This study documents this relationship for Bandon Bay for the first time, though it has been reported in other shallow water

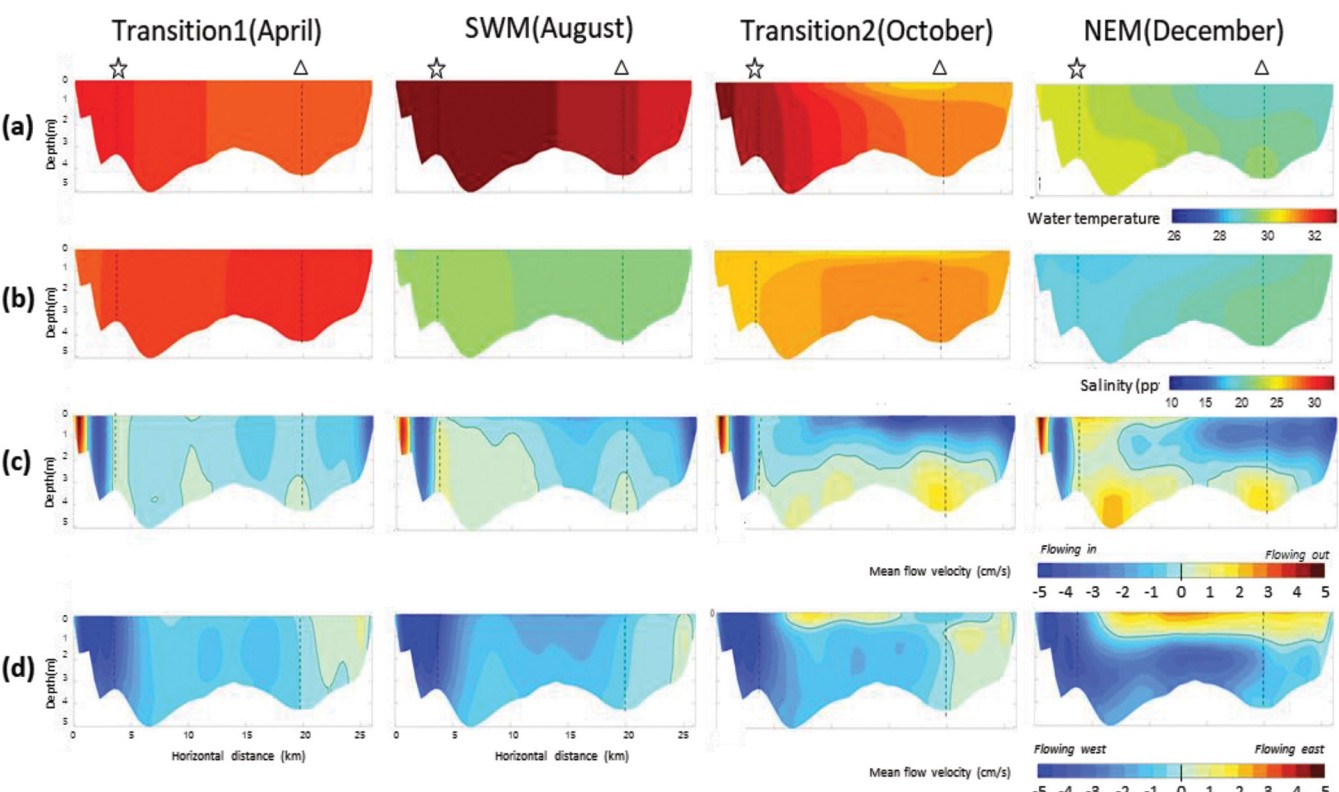

**Fig 7.** Distribution along water depth of monthly mean (a) salinity, (b) water temperature, (c) perpendicular flow velocity and (d) parallel flow velocity along the bay's mouth. Star and triangle with dashed line indicate positions of Sed Island and Prab Island.

bodies that receive similar substantial freshwater discharge and seasonal wind [e.g., 44,45]. Water temperature in Bandon Bay was higher when compared with the outer sea, regardless of the season (Fig 6B), and fell within the favorable range for primary production in marine environments [46,47]. Moreover, tidal currents were strong and provided good mixing of water in Bandon Bay in most months. This force sufficiently stirs fine sediment from the bottom throughout the water column, and possibly is important in delivering nutrients from sediment, which stimulate higher productivity of phytoplankton as food for aquatic animals in the bay [48,49]. Both phenomena are reflected by the fact that water in the bay is more turbid and has higher chlorophyll-*a* concentration when compared with the outer sea [50]. Stress from pollution is also less likely to occur since the retention time of water in this bay is very short (Fig 5). These factors may help explain why the living communities were rated as being under only moderate stress (based on W-statistics near zero) despite considerably high fishing pressure.

The W-statistic is a sensitive indicator of natural physical and biological disturbance as well as anthropogenic disturbance (such as pollution and fisheries) in both space and time [26,51]. The more stressed condition of the aquatic animal community reflected by the catches in sites 6 and 7 can be explained by their location at the eastern half of the bay. The hydrographic results clearly showed that salinity in the western half of the bay and at Sed Island was more stable when compared with the eastern half and Prab Island, where there is discharge from many canals and the Tapi River (Figs 6A and 7A). Moreover, although water temperature in Bandon Bay was always higher than the outer sea (Fig 6B), this difference was more pronounced at these two sites, which are in a shallower part of the bay than the others. Tweedley et al. [51] mentioned that environmental fluctuation would create more stress to crustaceans

and other macro-invertebrates (i.e., major catch items in traps) that have limited mobility compared to fishes. Higher W-statistic values for sites 3, 4 and 5 can be explained by strong tidal currents in the area, in particular during flood and ebb tides (Figs 5, 6C and 6D), and by strong flow (Fig 7C and 7D). Moreover, stable salinity would allow many marine fishes such as *Terapon* spp., being in the catches of trap. This is likely because these fishes undertake regular movements in synchrony with the tidal cycle to the area [52]. In terms of temporal variation, lower W-statistics during the first monsoon transition (i.e., April and May) and higher values in October and February could be partially explained by the current, which clearly differed between these periods. The strong currents in October and February, both at the surface and bottom (Fig 6C and 6C), as well as high parallel flow velocity (Fig 7D) would have caused resuspension of nutrients and then carried them along and across the bay time [53,54]. In contrast, these three forces were less pronounced during the first monsoon transition.

The assemblage patterns of the catches were highly related to seasonal variation, and hence to the hydrographic features in each season. The water temperatures associated with Cluster A were relatively low compared to the other two clusters (Figs 6B and 7B). Bacheler and Shertzer [55] showed that water temperature strongly influences trap catchability; higher temperature makes fishes and shellfishes more active and enhances the possibility of these animals being caught. This could explain why the number of taxa in the catch with %O greater than 5% in Cluster A was lower than in Clusters B and C. There is no clear hydrographic feature to explain differences between Clusters A1 and A2, but instead the relative numbers and weight of two brackish species (i.e., fish *T. oblongus* and crab *T. spinimana*) appeared to separate these groups of samples. Cluster B comprised samples largely from the second monsoon transition, i.e., October. During this time, the water was brackish to saline and relatively high in temperature, with stratification between surface and bottom (Fig 7A and 7B). Intrusion of seawater in the west side of the bay may have triggered more immigration of marine visitors such as cephalopods *Sepia* spp. and *Sepiella inermis* as well as marine demersal fishes such as *Paramonacanthus choirocephalus*, *Upeneus sulphureus* and *Terapon theraps*, for which %O was over 5% in this cluster [12,56]. Meanwhile, flushing by freshwater on the east right side of the bay kept the salinity relatively low. Although there was a clear difference in salinity between the west and east side of the bay (Fig 7A), there was no spatial difference in catch assemblages. This is likely because secondary freshwater fishes were not moving near the bottom and being caught by traps, whereas gillnets capture some secondary freshwater fishes when a high volume of freshwater is discharged into the Bay [14]. It is unfortunate that there were no samples from December, since it was expected to yield a different cluster showing high %O of secondary freshwater fishes, as experienced in the nearby Pak Panang Bay during the peak of the northeast monsoon (NEM) period, i.e., December to January [37,38]. Cluster C was composed of samples from late NEM to the first monsoon transition (i.e., from February to April), when water temperature was rising. This was also the main factor dividing cluster C1 (samples in February) and C2 (samples in March and April) (Figs 6B and 7B). It is also quite obvious that the mean current both near the surface and bottom were relatively lower for this group of samples than in other months of the year (Fig 6C and 6D), and this made the catch assemblages in cluster C different from the other two main clusters. Previous studies showed catches by traps in areas of weak current to be higher than those in strong current [57,58]. This could explain why the %O values for crabs in this cluster, in particular *Charybdis* spp. and *P. pelagicus*, were higher than in the other two clusters. The greater number of taxa with %O >5% in this cluster may be partially explained by low velocity, since it is considered as one of the key factors that shape the structure of demersal communities [59,60]. A strong current can create stress for aquatic animals, which cause them to seek for refuge and become less active, whereas calmer waters allow more activity and more chance for capture by trap nets [60,61].

## Conclusion

This study has demonstrated the significance of hydrographical features in shaping the aquatic animal community and consequently the catch composition of the trap fishery in a productive tropical bay. This fishery harvested almost 120 taxa of aquatic animals out of over 300 species known to inhabit Bandon Bay. Despite relatively high fishing pressure, the community was judged to be under moderate disturbance, which could reflect the high productivity of the bay itself. The composition of catches in traps was notably related to salinity, water temperature and currents within the bay, which also differed distinctly by season. These insights are essential for the implementation of sustainable resource management, in particular for areas with high fishing density and a high number of small-scale fishers dependent on the fishery. Moreover, this understanding also supports work on projecting trends in catches by this fishing gear in the face of anthropogenic stressors, such as, such as land reclamation along the coastline and climate changes, which would affect the hydrographical features of Bandon Bay.

## Supporting information

**S1 Fig. Hydrodynamic model validation against data from datalogger deployed near seabed from an observational station situated at the middle of the Bandon Bay in terms of salinity, water temperature and water level.**
(PDF)

**S2 Fig. Examples of the Abundance–Biomass Curve (ABC) and the estimated W-statistic from selected sampling events.** (a) example of ABC for undisturbed community with a positive sign W-statistic and (b) example of ABC for disturbed community with a negative sign W-statistic.
(PDF)

**S3 Fig. Near-surface distribution of monthly average salinity of Bandon Bay from the model.**
(PDF)

**S4 Fig. Near-surface distribution of monthly average water temperature of Bandon Bay from the model.**
(PDF)

**S5 Fig. Near-surface distribution of monthly average flow velocity from the Bandon Bay model.**
(PDF)

**S6 Fig. Modelled vertical distribution of monthly average salinity across Bandon Bay's mouth.**
(PDF)

**S7 Fig. Modelled vertical distribution of monthly average water temperature across Bandon Bay's mouth.**
(PDF)

**S8 Fig. Modelled vertical distribution of monthly average flow velocity across Bandon Bay's mouth.**
(PDF)

**S1 Table. Predicted occurrence probability (%) of each species in each cluster as calculated during the training process of SOM.**
(XLSX)

## Acknowledgments

This research was conducted to support the fishery improvement project of Thailand blue swimming crab (bottom gillnet/trap) in Suratthani Province. We are very grateful to all local fishers, whom collaborated with the project. Comments made by the reviewers have greatly improved the manuscript.

## Author Contributions

**Conceptualization:** Amonsak Sawusdee, Tanuspong Pokavanich, Tuantong Jutagate.

**Data curation:** Amonsak Sawusdee, Tanuspong Pokavanich, Sontaya Koolkalya, Jantira Rattanarat, Jenjira Kaewrat, Tuantong Jutagate.

**Formal analysis:** Amonsak Sawusdee, Tanuspong Pokavanich, Sontaya Koolkalya, Tuantong Jutagate.

**Funding acquisition:** Amonsak Sawusdee, Tuantong Jutagate.

**Supervision:** Tuantong Jutagate.

**Validation:** Tuantong Jutagate.

**Visualization:** Amonsak Sawusdee, Tanuspong Pokavanich, Jantira Rattanarat, Tuantong Jutagate.

**Writing – original draft:** Amonsak Sawusdee, Tanuspong Pokavanich, Sontaya Koolkalya, Jantira Rattanarat, Jenjira Kaewrat, Tuantong Jutagate.

**Writing – review & editing:** Amonsak Sawusdee, Tanuspong Pokavanich, Sontaya Koolkalya, Tuantong Jutagate.

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
