## [Decision Letter · Decision Letter 0]

24 Oct 2023

PONE-D-23-26740Catch assemblages in the small-scale trap fishery with relation to hydrographic features of a tropical bay in the Gulf of ThailandPLOS ONE

Dear Dr. Jutagate,

Thank you for submitting your manuscript to PLOS ONE. After careful consideration, we feel that it has merit but does not fully meet PLOS ONE’s publication criteria as it currently stands. Therefore, we invite you to submit a revised version of the manuscript that addresses the points raised during the review process.

We look forward to receiving your revised manuscript.

Kind regards,

Dharmendra Kumar Meena

Academic Editor

PLOS ONE

Journal Requirements:

"NO authors have competing interests"

5. Please upload a new copy of Figure 7 as the detail is not clear. Please follow the link for more information: " ext-link-type="uri" xlink:type="simple">https://blogs.plos.org/plos/2019/06/looking-good-tips-for-creating-your-plos-figures-graphics/"
" ext-link-type="uri" xlink:type="simple">https://blogs.plos.org/plos/2019/06/looking-good-tips-for-creating-your-plos-figures-graphics/"

6. We note that [Figures 1, 6 and S3-8] in your submission contain [map/satellite] images which may be copyrighted. All PLOS content is published under the Creative Commons Attribution License (CC BY 4.0), which means that the manuscript, images, and Supporting Information files will be freely available online, and any third party is permitted to access, download, copy, distribute, and use these materials in any way, even commercially, with proper attribution. For these reasons, we cannot publish previously copyrighted maps or satellite images created using proprietary data, such as Google software (Google Maps, Street View, and Earth). For more information, see our copyright guidelines: http://journals.plos.org/plosone/s/licenses-and-copyright.

a. You may seek permission from the original copyright holder of Figures 1, 6 and S3-8 to publish the content specifically under the CC BY 4.0 license.  

Additional Editor Comments:

Article is suggested to be revised

Reviewers' comments:

Reviewer's Responses to Questions

**Comments to the Author**

1. Is the manuscript technically sound, and do the data support the conclusions?

Reviewer #1: Yes

Reviewer #2: Yes

2. Has the statistical analysis been performed appropriately and rigorously? 

Reviewer #1: Yes

Reviewer #2: Yes

3. Have the authors made all data underlying the findings in their manuscript fully available?

Reviewer #1: Yes

Reviewer #2: Yes

4. Is the manuscript presented in an intelligible fashion and written in standard English?

Reviewer #1: Yes

Reviewer #2: Yes

5. Review Comments to the Author

Reviewer #1: This study was good by using the passive fishing method which is economically as well as environmentally beneficial by avoiding the pollution and cost which spend during the fishing operation. These methods are make the resources for long term viable or sustainable without the loss of biodiversity of flora and fauna.

Reviewer #2: The manuscript gives an insight into the effects of small-scale trap fishery on the community of aquatic animals and patterns of catch assemblage and association of hydrographic features of the bay with catch assemblage and community structure in the trap fishing ground. Trap fishery being a major contributor to the country’s fish production and the targeted species Portunus pelagicus being a commercially important species, this study holds significance to implement sustainable management measures.

Introduction is well written citing the details of the small scale fishery sector, fishery characteristics, the issues faced by SSF, and the importance of the Bandon Bay habitat. Objective of the work is clear and specific.

The authors have done a commendable work in assessing the spatio-temporal variation of catch assemblage in the trap fishery and the hydrographic association. Results and discussions justify the objective of the work and conclusion highlights the importance of this study.

In figure 1, the 14th sampling site is not indicated.

Line 124: the name of cyclone is Pabuk

Line 214: correct the sampling year

6. PLOS authors have the option to publish the peer review history of their article (what does this mean?). If published, this will include your full peer review and any attached files.

Reviewer #1: **Yes: **Dr. Veerendra Singh

Reviewer #2: No

---

## [Author Response · Author response to Decision Letter 0]

3 Dec 2023

Response to reviewers

Reviewer#1

Thank you very much for the general comment by Reviewer #1 and there is no specific comment made by Reviewer #1.

Reviewer#2

Thank you very much for the general comment by Reviewer #2. I

1. In figure 1, the 14th sampling site is not indicated.

Response: Figure 1 was already revised and made correction.

2. Line 124: the name of cyclone is Pabuk

Response: The name of the cyclone in text was already corrected.

3. Line 214: correct the sampling year

Response: The year in caption of Table 1 was already corrected, i.e., 2018

END OF RESPONSE

---

## [Editor Report · Decision Letter 1]

7 Dec 2023

Catch assemblages in the small-scale trap fishery with relation to hydrographic features of a tropical bay in the Gulf of Thailand

PONE-D-23-26740R1

Dear Dr. Jutagate

We’re pleased to inform you that your manuscript has been judged scientifically suitable for publication and will be formally accepted for publication once it meets all outstanding technical requirements.

Kind regards,

Dharmendra Kumar Meena

Academic Editor

PLOS ONE

Additional Editor Comments (optional):

The article can now be accepted.

regards

D K Meena
---

## [Editor Report · Acceptance letter]

11 Dec 2023

PONE-D-23-26740R1 

Catch assemblages in the small-scale trap fishery with relation to hydrographic features of a tropical bay in the Gulf of Thailand 

Dear Dr. Jutagate:

I'm pleased to inform you that your manuscript has been deemed suitable for publication in PLOS ONE. Congratulations! Your manuscript is now with our production department. 

Kind regards, 

on behalf of

Dr. Dharmendra Kumar Meena 

Academic Editor

PLOS ONE